# Characterization of the kinetic cycle of an ABC transporter by single-molecule and cryo-EM analyses

Ling Wang[1†], Zachary Lee Johnson[2†], Michael R Wasserman[1†], Jesper Levring[2†], Jue Chen[2,3]*, Shixin Liu[1]*

[1]Laboratory of Nanoscale Biophysics and Biochemistry, The Rockefeller University, New York, United States; [2]Laboratory of Membrane Biology and Biophysics, The Rockefeller University, New York, United States; [3]Howard Hughes Medical Institute, The Rockefeller University, New York, United States

**Abstract** ATP-binding cassette (ABC) transporters are molecular pumps ubiquitous across all kingdoms of life. While their structures have been widely reported, the kinetics governing their transport cycles remain largely unexplored. Multidrug resistance protein 1 (MRP1) is an ABC exporter that extrudes a variety of chemotherapeutic agents and native substrates. Previously, the structures of MRP1 were determined in an inward-facing (IF) or outward-facing (OF) conformation. Here, we used single-molecule fluorescence spectroscopy to track the conformational changes of bovine MRP1 (bMRP1) in real time. We also determined the structure of bMRP1 under active turnover conditions. Our results show that substrate stimulates ATP hydrolysis by accelerating the IF-to-OF transition. The rate-limiting step of the transport cycle is the dissociation of the nucleotide-binding-domain dimer, while ATP hydrolysis per se does not reset MRP1 to the resting state. The combination of structural and kinetic data illustrates how different conformations of MRP1 are temporally linked and how substrate and ATP alter protein dynamics to achieve active transport.

*For correspondence:
juechen@rockefeller.edu (JC);
shixinliu@rockefeller.edu (SL)

[†]These authors contributed equally to this work

Competing interests: The authors declare that no competing interests exist.

## Introduction

Multidrug resistance protein 1 (MRP1) is an ATP-binding cassette (ABC) transporter that harnesses the energy of ATP to extrude substrates from the cytosol to the extracellular space (*Cole, 2014a*). Native substrates of MRP1 include a variety of antioxidants, pro-inflammatory molecules, and hormones (*Cole and Deeley, 2006*; *Deeley and Cole, 2006*; *Deeley et al., 2006*; *Leslie et al., 2005*). MRP1 also transports a number of chemotherapeutic agents, thereby conferring drug resistance in acute myeloblastic and lymphoblastic leukemia, non-small-cell lung cancer, prostate cancer, breast cancer, and neuroblastoma (*Berger et al., 2005*; *Cole, 2014b*; *Filipits et al., 2005*; *Haber et al., 2006*; *Winter et al., 2013*; *Zalcberg et al., 2000*).

MRP1 is a single polypeptide comprising three transmembrane domains (TMD0, TMD1, and TMD2) and two cytosolic nucleotide-binding domains (NBD1 and NBD2). Structures of bovine MRP1 (bMRP1) have been determined by electron cryo-microscopy (cryo-EM) in three functional states (*Johnson and Chen, 2017*; *Johnson and Chen, 2018*): an apo form in the absence of substrate and ATP, a complex with the native substrate leukotriene $C_4$ ($LTC_4$) in the absence of ATP, and a structure of the hydrolysis-deficient E1454Q mutant determined in the presence of both $LTC_4$ and ATP (*Figure 1A*). These structures, in accord with decades of functional analysis (*Cole, 2014a*) bring about the following understanding of the transport cycle. In the absence of ATP, MRP1 adopts an inward-facing (IF) conformation, in which the two NBDs are widely separated and the translocation pathway is open to the cytoplasm. Binding of $LTC_4$ at the center of the membrane, between TMD1

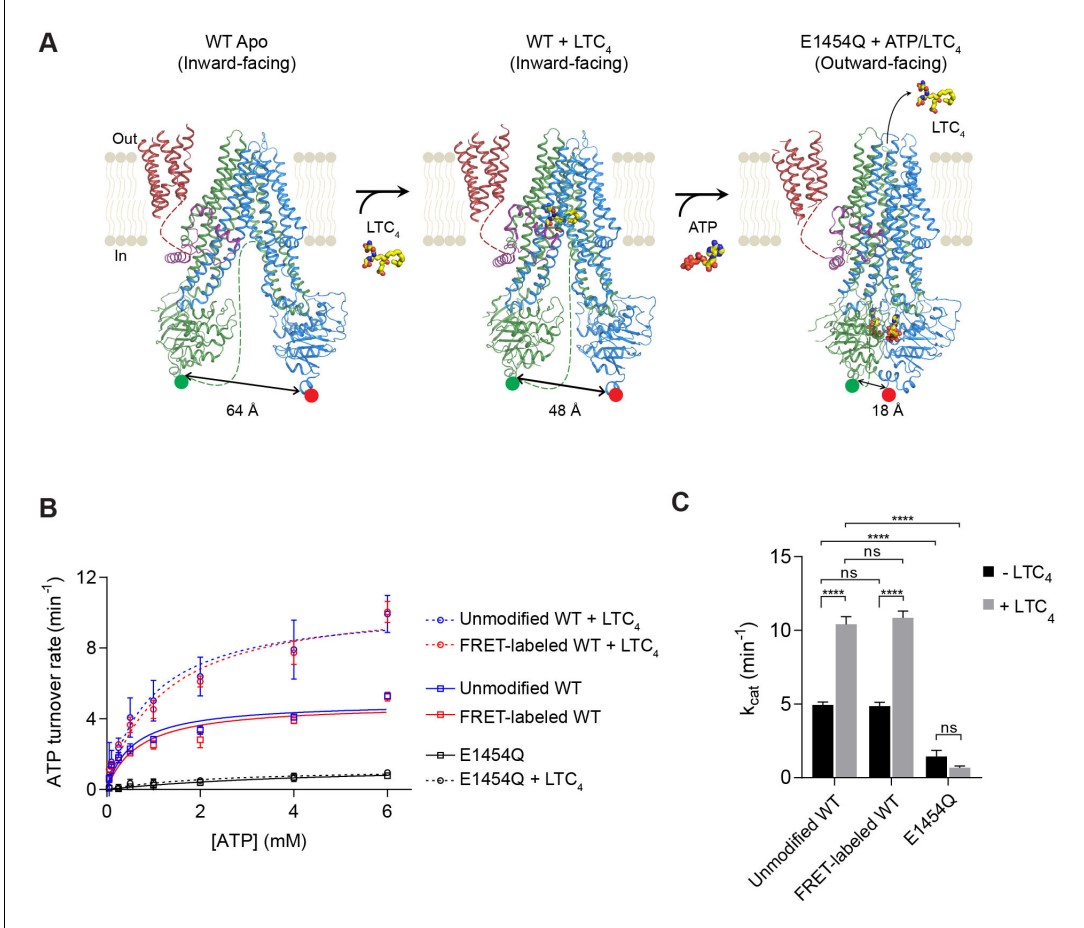

**Figure 1.** Structure-guided smFRET design for probing MRP1 dynamics. (**A**) Structures of bMRP1 captured in ligand-free (left), LTC$_4$-bound (middle), and ATP-bound (right) conformations (PDB accession numbers: 5UJ9, 5UJA, and 6BHU). The positions of tag insertions for site-specific labeling are highlighted in green (FRET donor; peptide sequence substituted following NBD1 residue 867) and red (FRET acceptor; peptide sequence inserted at the C-terminus following NBD2 residue 1530). The distances shown correspond to the separations between residues 867 and 1530 rather than the inter-probe distances. TMD0 is shown in red, the lasso motif in purple, TMD1/NBD1 in green, and TMD2/NBD2 in blue. Flexible linkers not observed in the cryo-EM maps are represented by dotted lines. (**B**) ATPase activity of unmodified wild-type (WT), FRET-labeled WT, and E1454Q mutant bMRP1, either without or with LTC$_4$ (10 μM). Data are represented as mean ±95% confidence intervals (from 3 to 6 independent measurements) and fitted to Michaelis-Menten equations. (**C**) Catalytic constants for ATP turnover by unmodified WT, FRET-labeled WT, and E1454Q bMRP1 in the absence and presence of 10 μM LTC$_4$. Data are represented as mean ± SEM. Comparisons were made by one-way ANOVA (****p<0.0001; ns, not significant). The residual ATP turnover seen in the E1454Q sample did not respond to LTC$_4$ stimulation, and was thus most likely due to spontaneous ATP hydrolysis independent of MRP1 activity.

The online version of this article includes the following figure supplement(s) for figure 1:

**Figure supplement 1.** Site-specific labeling and purification of MRP1.

and TMD2, brings the two halves of the transporter closer together. Upon binding of ATP, the transporter adopts its fully NBD-closed configuration concurrent with opening of the translocation pathway to the outside. Meanwhile, the LTC$_4$-binding pocket becomes deformed, no longer competent to bind substrate. In this outward-facing (OF) conformation, two ATP molecules are occluded at the NBD dimer interface: one in the catalytically inactive, degenerate site and the other in the active consensus site, poised for hydrolysis (*Figure 1A*).

While a wealth of structural characterizations of ABC transporters provide atomistic details of these proteins in specific conformations (*Oldham et al., 2008*; *Srikant and Gaudet, 2019*), they do not inform us of how these structural snapshots are temporally linked. On the other hand, single-molecule techniques are well suited for tracking dynamic processes and have been applied to study a number of membrane proteins (*Akyuz et al., 2013*; *Dyla et al., 2017*; *Goudsmits et al., 2017*;

*Husada et al., 2018*; *Wang et al., 2016*; *Zhao et al., 2010*). In this study, we took a combined single-molecule and structural approach to investigate the kinetic mechanism that governs the transport cycle of MRP1. Based on prior structural information, we designed single-molecule fluorescence resonance energy transfer (smFRET) experiments to track the conformational changes of bMRP1 in real time. Guided by the smFRET results, we then used cryo-EM to capture the most abundant structural configuration of wild-type (WT) MRP1 during active turnover. The synergy between the smFRET and cryo-EM studies has enabled us to determine the rate-limiting mechanism of MRP1 and how substrate and ATP modulate the kinetics of this important drug transporter.

## Results

### smFRET design

Cryo-EM structures indicate that in the MRP1 transport cycle the two NBDs undergo large motions of association and dissociation (*Figure 1A*). To monitor the conformational dynamics of bMRP1 at the single-molecule level, we introduced two fluorophore labeling sites at the distal ends of NBD1 and NBD2. A 12-residue S6 peptide replaced residues 868–879, and a 12-residue A1 peptide was added to the C-terminus following residue V1530. Cy3 (donor) and LD655 (acceptor) fluorophores were orthogonally conjugated to bMRP1 using Sfp and AcpS synthases, respectively (*Figure 1—figure supplement 1*). In addition, a $His_{10}$-tag was added at the N-terminus of TMD0 for surface immobilization. In solution, the fluorescently labeled bMRP1 hydrolyzed ATP at rates virtually identical to those of the unmodified WT protein (*Figure 1B and C*), indicating that insertion of the peptide tags and incorporation of the fluorophores did not alter the kinetics of bMRP1.

### Conformational distributions of MRP1

Using total-internal-reflection fluorescence (TIRF) microscopy, we measured the steady-state distribution of FRET efficiency (*E*) for WT bMRP1 under five different conditions (*Figure 2A*). In the absence of ATP and substrate (apo), the FRET histogram showed a broad distribution of *E* values spanning from 0.2 to 0.9. Saturating concentrations of $LTC_4$ (10 μM) or ATP (5 mM) shifted the FRET distribution towards higher *E* values. When both $LTC_4$ and ATP were present, a predominant high FRET peak emerged. Addition of ATP and $LTC_4$ together with orthovanadate (Vi), a hydrolysis transition-state analogue, further promoted the high FRET state (*Figure 2A*). We then used the same labeling strategy to attach the FRET dye pair to the catalytically inactive E1454Q mutant bMRP1. The FRET distribution for the E1454Q mutant in the presence of ATP and $LTC_4$ was also dominated by a high FRET state (*Figure 2A*).

To quantitatively characterize the FRET distributions, we applied the empirical Bayesian method implemented in ebFRET (*van de Meent et al., 2014*) and identified a common set of five FRET states, with mean *E* values of 0.23, 0.42, 0.63, 0.80, and 0.92. We then used a hidden Markov modeling (HMM) algorithm (*Qin, 2004*) to idealize the smFRET time trajectories to these discrete states. The FRET distributions obtained under all above experimental conditions can be described as combinations of the five states (*Figure 2A*, lower panel). A four-state model is insufficient to describe transitions observed in the FRET trajectories (*Figure 2—figure supplement 1*). Increasing the number of states to six only resulted in an additional unpopulated state, indicative of overfitting (*Figure 2—figure supplement 1*).

Next we sought to correlate these five FRET states with existing structural information. To identify the $LTC_4$-bound conformation, we collected smFRET data with a series of $LTC_4$ concentrations and analyzed the FRET distributions using the five-state model (*Figure 2—figure supplement 2A*). The occupancy of the *E* = 0.80 state increased monotonically as the $LTC_4$ concentration was raised, whereas those of the other states did not (*Figure 2B* and *Figure 2—figure supplement 2C*), consistent with the peak position of 0.8 from the overall FRET distribution collected at saturating $LTC_4$ (*Figure 2A*). The half-maximal effective concentration ($EC_{50}$) of $LTC_4$ for the *E* = 0.80 state occupancy is 0.32 ± 0.17 μM, in agreement with the $LTC_4$ concentration required for half-maximal stimulation of ATP hydrolysis measured in a bulk ATPase assay (0.35 ± 0.09 μM) (*Johnson and Chen, 2017*). Therefore we rationalized that the *E* = 0.80 state (termed $IF_1$) corresponds to the inward-facing cryo-EM structure determined in the presence of saturating $LTC_4$ (*Figure 1A*, middle).

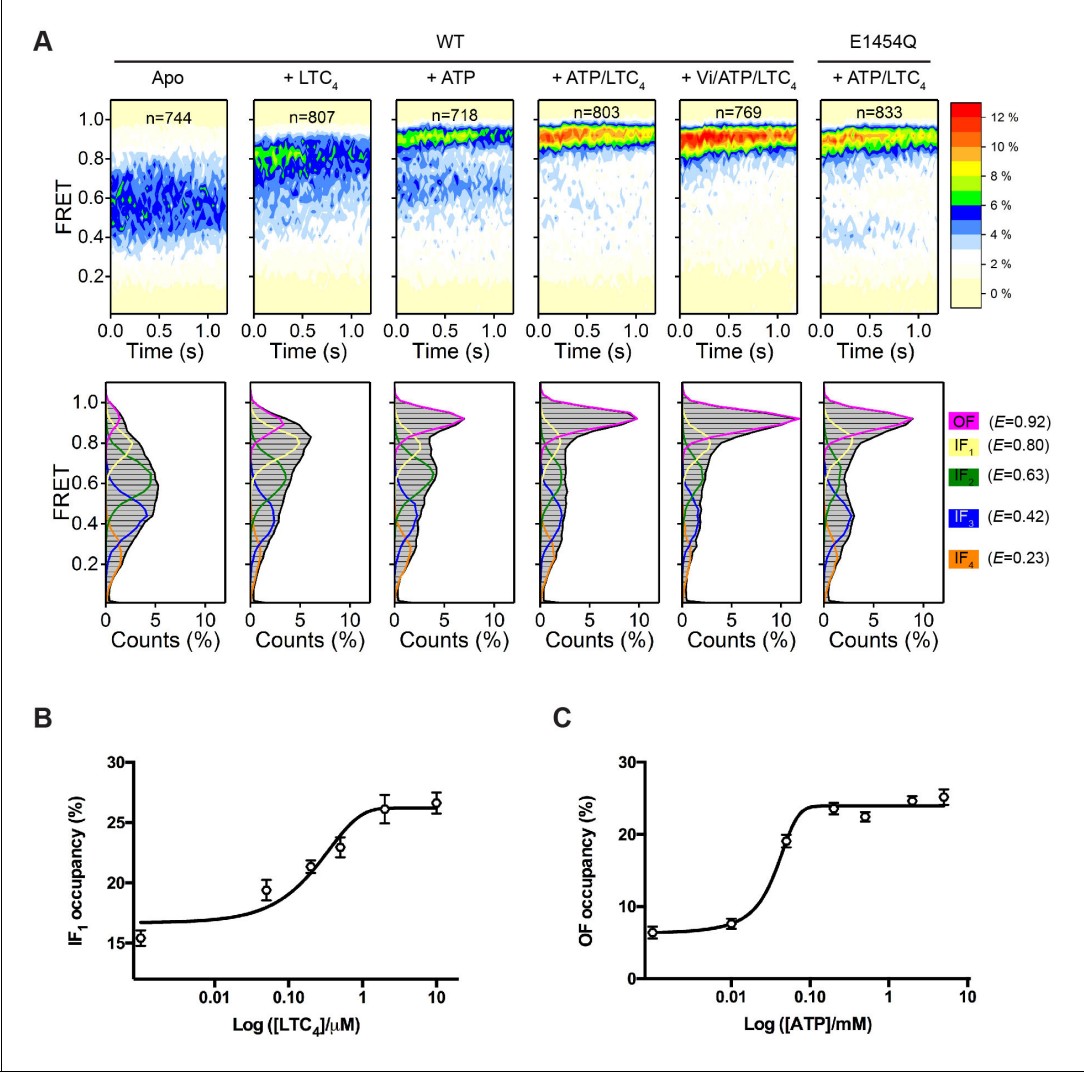

**Figure 2.** Conformational landscapes of MRP1 revealed by smFRET. (**A**) Contour plots (top) and histograms (bottom) of FRET distributions obtained with WT bMRP1 in the following conditions (from left to right): apo, + LTC$_4$ (10 μM), + ATP (5 mM), + ATP/LTC$_4$ (5 mM/10 μM), + Vi/ATP/LTC$_4$ (1 mM/5 mM/10 μM). Shown in the right column are data for the E1454Q mutant in the presence of ATP/LTC$_4$ (5 mM/10 μM). The time-dependent changes in the contour plots were due to fluorophore photobleaching, which depopulated FRET-active molecules over time. Time points after photobleaching were excluded from subsequent analysis. The histograms represent the cumulative FRET distributions over the entire 1.25 s time window. Overlaid on the histograms are fitted distributions by the five-state model with mean FRET values of 0.92 (magenta, OF), 0.80 (yellow, IF$_1$), 0.63 (green, IF$_2$), 0.42 (blue, IF$_3$), and 0.23 (orange, IF$_4$). n denotes the number of molecules analyzed. (**B**) Relative occupancy of the IF$_1$ state in the presence of increasing concentrations of LTC$_4$. Data are fitted to a dose-response function with the Hill equation, yielding an EC$_{50}$ of 0.32 ± 0.17 μM. (**C**) Relative occupancy of the OF state in the presence of increasing concentrations of ATP. Data are fitted to a dose-response function with the Hill equation, yielding an EC$_{50}$ of 0.05 ± 0.02 mM. Data are represented as mean ± SEM.

The online version of this article includes the following figure supplement(s) for figure 2:

**Figure supplement 1.** Determination of model parameters for idealizing smFRET trajectories.

**Figure supplement 2.** Dependence of the conformational distribution of MRP1 on substrate and ATP.

The highest FRET state (*E* = 0.92) indicates a conformation in which the NBDs are in closer proximity than that of the LTC$_4$-bound structure. Its occupancy increased as a function of ATP concentration, but those of the lower FRET states did not (*Figure 2C*, *Figure 2—figure supplement 2B and D*), consistent with visual inspection of the overall FRET distribution (*Figure 2A*). Moreover, the E1454Q mutant predominately occupied the *E* = 0.92 state at saturating concentrations of ATP and LTC$_4$ (*Figure 2A*), similar to the condition under which the cryo-EM structure of the outward-facing

conformation was determined (*Johnson and Chen, 2018*). These observations support a direct correspondence between the *E* = 0.92 state and the OF conformation.

The three lowest FRET states (i.e., *E* = 0.63, 0.42, and 0.23) indicate conformations in which the NBDs are further separated than in the LTC$_4$-bound structure, and thus are presumably ligand-free and inward-facing (termed IF$_2$, IF$_3$, and IF$_4$, respectively). The presence of multiple apo states underscores the conformational flexibility of MRP1 in the absence of ligand, a well-documented characteristic of many ABC transporters (*Husada et al., 2018*; *Timachi et al., 2017*; *Ward et al., 2007*; *Ward et al., 2013*). In the cryo-EM structure of apo bMRP1, the local resolution of the NBDs was poorer compared to other regions, suggesting that the NBDs are relatively mobile (*Johnson and Chen, 2017*).

From the idealized smFRET trajectories we extracted the lifetimes of each state and transition frequencies between each pair of states. In the absence of ATP, the vast majority of transitions

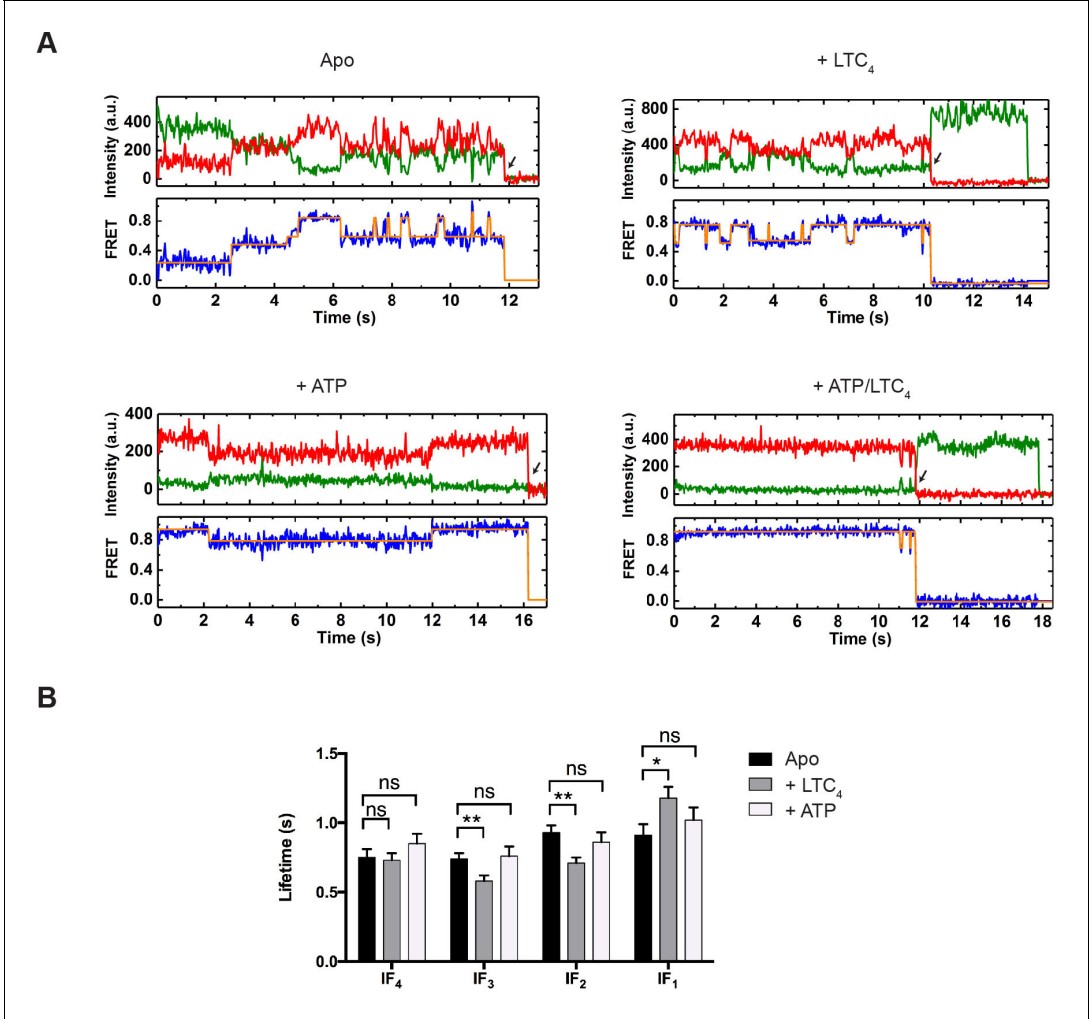

**Figure 3.** Real-time dynamics of MRP1 under steady state. (**A**) Representative single-molecule donor (green) and acceptor (red) fluorescence trajectories, and the corresponding FRET trajectories (blue) obtained at a frame rate of 25 ms. Idealized FRET states are overlaid in orange lines. Arrows indicate fluorophore photobleaching events, after which the data were excluded from further analysis. The following conditions were analyzed: apo, + LTC$_4$ (10 µM), + ATP (5 mM), and + ATP/LTC$_4$ (5 mM/10 µM). (**B**) Average lifetimes of each IF state under different conditions. Data are represented as mean ± SEM.

The online version of this article includes the following figure supplement(s) for figure 3:

**Figure supplement 1.** Additional representative fluorescence and FRET trajectories obtained at 25 ms time resolution for the indicated conditions.
**Figure supplement 2.** Transition density plots displaying the distributions of FRET values before (x-axis) and after (y-axis) each transition identified in the idealized FRET traces.

occurred among the four IF states ($IF_1$, $IF_2$, $IF_3$, and $IF_4$) (**Figure 3A**, **Figure 3—figure supplements 1** and **2**). The lifetime of each IF state was on average around 1 s (**Figure 3B**). Addition of $LTC_4$ increased the lifetime of $IF_1$ by about 30%, lending further support to its assignment as the substrate-bound state, whereas ATP did not produce statistically significant changes in the lifetime of any IF state (**Figure 3B**).

## Determinants of IF-to-OF transitions

Estimation of the lifetime of the OF state was not possible at a temporal resolution of 25 ms. At this time resolution and corresponding laser power, molecules often remained in the OF state in the presence of ATP until fluorophore photobleaching occurred (**Figure 3—figure supplement 1**). In order to capture the complete duration of the OF state, we lowered the time resolution to 300 ms and decreased the laser power. Under these imaging conditions, the average lifetime of the fluorophores was approximately 100 s. This allowed us to observe multiple cycles of IF-OF interconversions, albeit with the drawback of missing some fast transitions.

To better understand the mechanism of IF-to-OF transitions, we performed real-time perturbation experiments, in which ATP was delivered into the sample chamber containing apo MRP1. Upon examining individual smFRET trajectories, we observed two distinct populations of MRP1: one population showed clear transitions from the IF to the OF state after ATP injection and then cycled between IF and OF states (**Figure 4A**), while the other population only transitioned among different IF states and never reached the OF state during the observation period (**Figure 4—figure supplement 1A**). Molecules in the latter group were presumed to be biochemically inactive, as ATP hydrolysis only occurs in the NBD-dimerized OF conformation. Prior to ATP injection, both groups of molecules were predominantly in the IF conformations but exhibited distinct FRET distributions (**Figure 4—figure supplement 1B**). Addition of ATP substantially increased the occupancy of the OF state within the active group (**Figure 4B** and **Figure 4—figure supplement 1B**), whereas the FRET distribution of the inactive group did not change upon ATP injection (**Figure 4—figure supplement 1B**). These inactive molecules must also exist in the steady-state experiments described in **Figure 2**. However, due to the short observation window with a 25 ms frame rate it was not possible to identify these molecules.

Using the trajectories of active molecules only, we measured the wait time ($t_{wait}$) from the point of ATP injection to the onset of the first OF state (**Figure 4A**). This duration is a compound function of several events including ATP binding and NBD dimerization. Increasing the ATP concentration from 10 µM to 5 mM shortened $t_{wait}$ by approximately twofold (**Figure 4C**), consistent with the understanding that ATP binding promotes NBD dimerization. The presence of $LTC_4$ reduced $t_{wait}$ at both limiting and saturating ATP concentrations (**Figure 4C**), indicating that substrate accelerates the transition from the IF to OF state. Moreover, we found that the vast majority of transitions into the OF state originated from $IF_1$ or $IF_2$ (**Figure 4—figure supplement 2A**), suggesting that the substrate-bound configuration $IF_1$ is not an obligatory intermediate en route to the OF state.

After the active MRP1 molecules made the first IF-to-OF transition, they continued to cycle between IF and OF states (**Figure 4A** and **Figure 4—figure supplement 2B**). Given that the 300 ms time resolution obscured a fraction of fast transitions among different IF states, we used the composite time that molecules spent in the IF states before converting into the OF state ($t_{IF}$; **Figure 4A**) to describe the kinetics of IF-to-OF transitions. As expected from the notion that ATP drives the IF-to-OF isomerization, increasing the ATP concentration shortened $t_{IF}$ (**Figure 4D**). $LTC_4$ further reduced the average $t_{IF}$ by approximately threefold (**Figure 4D**).

## Determinants of OF-to-IF transitions

To investigate how the reverse transition (i.e., from the OF to IF state) is influenced by ATP and $LTC_4$, we measured the lifetime of OF states ($t_{OF}$; **Figure 4A**). In control experiments where the molecules were incubated with buffer or $LTC_4$ alone in the absence of ATP, spontaneous transitions into the OF state were occasionally observed (**Figure 5A**), but the resultant OF states on average only lasted a few seconds (**Figure 5B** and **Figure 5—figure supplement 1**). In comparison, the average OF lifetime measured in the presence of saturating ATP was longer than 20 s (**Figure 5B** and **Figure 5—figure supplement 1**), indicating that ATP stabilizes the OF conformation.

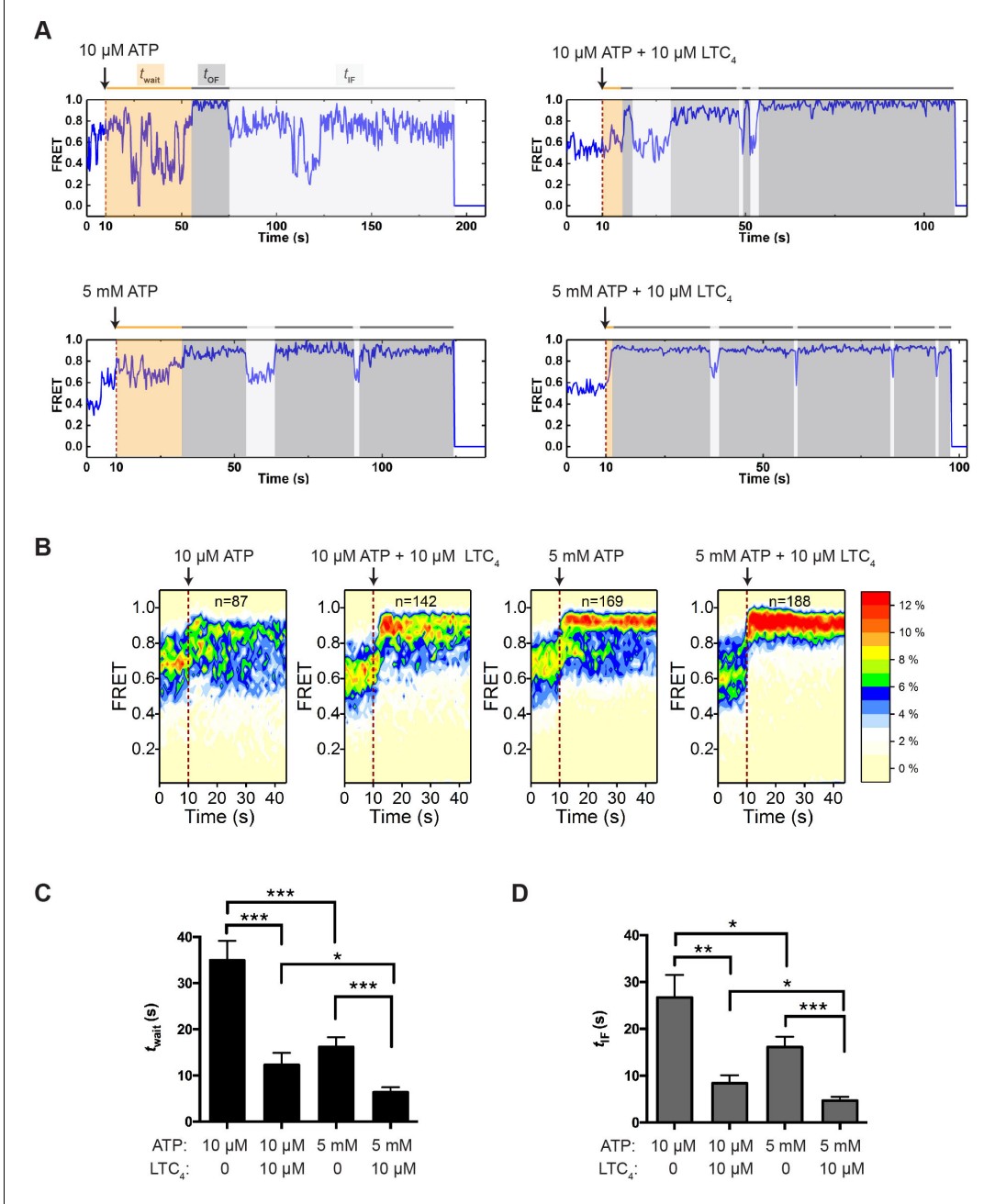

**Figure 4.** Transitions between IF and OF states from perturbation experiments. (A) Representative smFRET trajectories of active molecules from the perturbation experiments obtained at a frame rate of 300 ms. A limiting (10 μM) or saturating (5 mM) concentration of ATP with or without LTC$_4$ (10 μM) was injected into the imaging chamber at 10 s (dashed lines). The wait time until the onset of the first OF state ($t_{wait}$) and the lifetime of the subsequent OF states ($t_{OF}$) and IF states ($t_{IF}$) are shaded in orange, dark gray, and light gray, respectively. (B) Contour plots of smFRET trajectories for active molecules under each perturbation condition. n denotes the number of active molecules analyzed. (C) Average $t_{wait}$ for different perturbation conditions. (D) Average $t_{IF}$ for different perturbation conditions. The values have been corrected for the photobleaching rate. Data are represented as mean ± SEM.

The online version of this article includes the following figure supplement(s) for figure 4:

**Figure supplement 1.** Differentiation between inactive and active molecules.

**Figure supplement 2.** Characteristics of IF-to-OF transitions under different conditions.

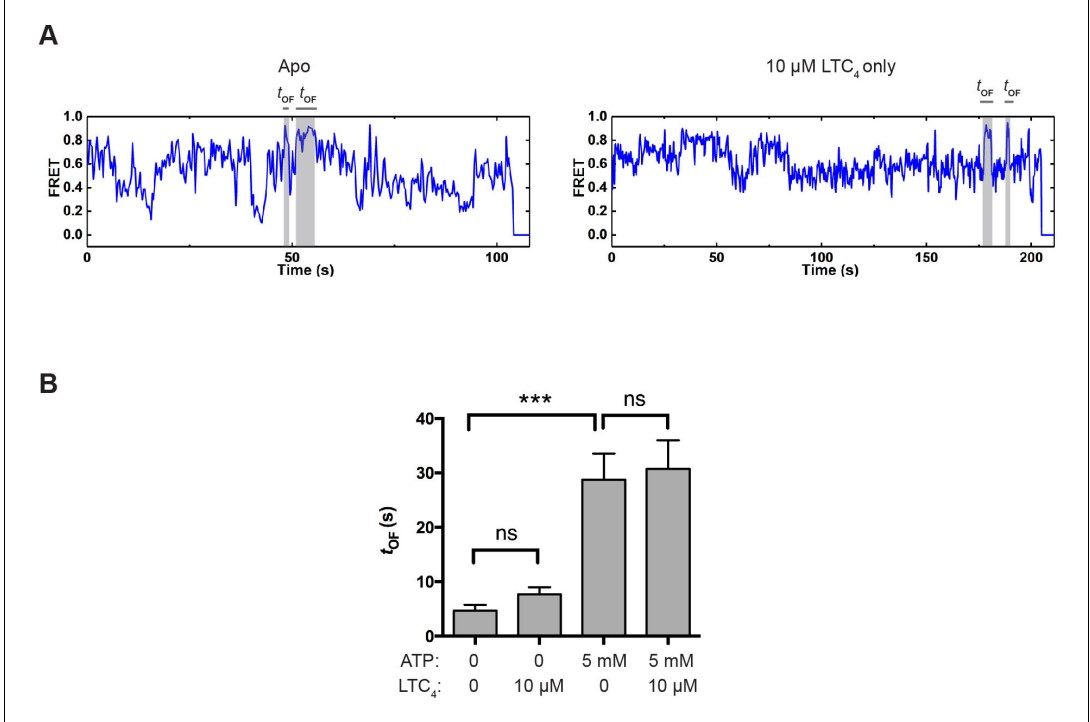

**Figure 5.** OF state lifetime under different conditions. (**A**) Example smFRET trajectories for apo and LTC₄-only conditions showing spontaneous transitions into the OF state independent of ATP. (**B**) Average lifetime of the OF states under different conditions. The values have been corrected for the photobleaching rate. Data are represented as mean ± SEM.

The online version of this article includes the following figure supplement(s) for figure 5:

**Figure supplement 1.** Histograms of OF state lifetimes for different conditions and their fit to single-exponential decay functions.

$LTC_4$ had only a minor effect on the kinetics of OF-to-IF transitions (*Figure 5B* and *Table 1*), consistent with the structural observation that $LTC_4$ is readily released in the ATP-bound OF conformation prior to ATP hydrolysis (*Johnson and Chen, 2018*), hence not affecting the subsequent isomerization back to the IF state.

When saturating concentrations of ATP and $LTC_4$ were present, the transition from the OF to the IF conformation was much slower than the IF-to-OF transition ($t_{OF}$ vs. $t_{IF}$, *Table 1*), also evident from the single-molecule trajectories (*Figure 4A* and *Figure 4—figure supplement 2B*). Therefore, it is likely that under physiological conditions where ATP is saturating, MRP1 spends the majority of its time in the OF conformation. The total cycle time measured from the single-molecule data ($t_{OF} + t_{IF}$, *Table 1*) is 4–6 times longer than that deduced from the bulk ATPase assay (*Figure 1C*). A number of factors could contribute to this difference. The temperature differed between the two assays (23°C for smFRET vs. 30°C for bulk assay). It is also possible that surface immobilization of MRP1 may

**Table 1.** Kinetics of the transitions between IF and OF conformations

| Condition | $t_{IF}$ (s) | $t_{OF}$ (s) |
| --- | --- | --- |
| WT, 5 mM ATP | 16.1 ± 2.2 | 28.8 ± 4.8 |
| WT, 5 mM ATP + 10 µM LTC₄ | 4.7 ± 0.8 | 30.8 ± 5.2 |
| E1454Q, 5 mM ATP + 10 µM LTC₄ | 7.7 ± 1.5 | 31.7 ± 5.5 |

Shown are the average lifetimes (mean ± SEM) of the composite IF state and the OF state for WT and E1454Q MRP1 with indicated ATP and substrate concentrations. The effect of dye photobleaching on the apparent IF/OF lifetime has been corrected for.

affect its activity. Finally, some fast transitions between IF and OF states may escape detection at the 300 ms time resolution of smFRET measurements.

## Cryo-EM structure of MRP1 under active turnover conditions

Next we pursued cryo-EM studies to further discern the rate-limiting step in the MRP1 transport cycle. Based on the smFRET data (*Figures 2A* and *4A*), we anticipated that most particles would adopt an OF conformation in the presence of saturating $LTC_4$ and ATP. Further, if ATP hydrolysis were rate limiting, the predominant species would be the pre-hydrolysis state with ATP molecules bound at both ATPase sites. Otherwise, we would expect to observe a dominant structure in which ATP in the catalytically competent, consensus site has already been hydrolyzed.

To prepare the sample for cryo-EM, WT bMRP1 (30 μM) was incubated with 80 μM $LTC_4$ and 10 mM ATP-$Mg^{2+}$ on ice for 10 min before plunge freezing in liquid ethane. If the sample had been prepared at room temperature, approximately 3 mM ATP would have been hydrolyzed during this incubation (*Figure 1C*). On ice, we expect a slower turnover rate and thus even less depletion of ATP. This approximates an 'active turnover' condition where ATP hydrolysis has reached steady state.

We first used all 1,143,729 particles from the cryo-EM dataset to calculate a 3.4 Å reconstruction that represents the dominant structure under the active turnover condition (*Figure 6—figure supplement 1* and *Table 2*). Consistent with the single-molecule data, the resulting map shows an NBD-dimerized conformation, with an ATP molecule in the degenerate site and an ADP molecule in the consensus site (*Figure 6—figure supplement 2A and B*). To improve the resolution of the structure, we also carried out 3D classification and obtained a higher quality map at 3.2 Å from a subset of the particles (*Figure 6—figure supplements 1* and *2*), which unambiguously shows an ADP molecule at the consensus site (*Figure 6*). This structure strongly suggests that ATP hydrolysis is a fast step in the transport cycle, whereas dissociation of the NBD dimer constitutes a kinetic bottleneck, thereby limiting the rate of the overall cycle.

Other than ADP occupying the consensus site in place of ATP, the overall structure of this post-hydrolysis conformation is essentially identical to that of the pre-hydrolytic structure solved with the hydrolysis-deficient E1454Q mutant (*Johnson and Chen, 2018*; *Figure 6A and B*). In both structures, the intracellular gate is closed and the substrate-binding site is pulled apart such that $LTC_4$ is no longer bound. Thus, for MRP1, ATP hydrolysis per se and the subsequent release of inorganic phosphate ($P_i$) do not induce conformational rearrangements.

## Discussion

Previously, using cryo-EM and mutagenesis, we captured three snapshots of MRP1, demonstrating the conformational changes induced by $LTC_4$ and ATP (*Johnson and Chen, 2017*; *Johnson and Chen, 2018*). Guided by these structures, here we employed single-molecule fluorescence spectroscopy to characterize the dynamics of MRP1 to understand the connectivity of these conformations. The kinetic results from these single-molecule experiments prompted us to solve the dominant MRP1 structure under active turnover conditions, which revealed the rate-limiting mechanism of the MRP1 transport cycle. Such a synergistic approach enabled us to obtain new insights into the kinetic cycle of MRP1, depicted in *Figure 7*.

By monitoring the separation of the two NBDs, we showed that MRP1 is intrinsically dynamic: in the absence of any ligand, MRP1 transitions among multiple IF conformations. It also spontaneously accesses the OF conformation, albeit with lower frequency and shorter lifetime in the absence of nucleotides. Similarly, a recent smFRET study showed that the *E. coli* peptide transporter McjD also samples an NBD-dimerized state without nucleotides (*Husada et al., 2018*). Whether all of these ground-state conformations are functionally important requires further investigation.

When ATP is present, as in physiological conditions, the MRP1 transport cycle is a nonequilibrium process coupled to ATP hydrolysis. ATP binding accelerates the IF-to-OF transition ($k_1$) and slows down the reverse OF-to-IF transition ($k_{-1}$) by stabilizing the OF conformation. Substrate, such as $LTC_4$, stimulates the ATPase activity through acceleration of the IF-to-OF transition ($k_1$), but not ATP hydrolysis ($k_2$) or the subsequent OF-to-IF transition ($k_3$). In this model, the same high FRET value ($E = 0.92$) is produced by two distinct NBD-dimerized states: the pre-hydrolytic state in which two ATP molecules are bound and an asymmetric post-hydrolytic state bound with one ATP and one ADP. Still, the lifetime distributions of the $E = 0.92$ state can be fit with single-exponential decay

**Table 2.** Summary of EM data and structure refinement statistics

**Data collection**

| | |
|---|---|
| Microscope | Titan krios (FEI) |
| Voltage (kV) | 300 |
| Detector | K2 Summit (Gatan) |
| Pixel size (Å) | 1.03 |
| Defocus range (μm) | 0.7 to 2.4 |
| Movies | 3604 |
| Frames/movie | 50 |
| Dose rate (electrons/pixel/s) | 8.0 |
| Total dose (electrons/Å$^2$) | 75 |
| Number of particles | 1,143,729 |
| **Model composition** | |
| Non-hydrogen atoms | 9684 |
| Protein residues | 1210 |
| Lipids/Detergents/Ligands | 3 CHS/1 ATP/1 ADP/2 Mg$^{2+}$ |
| **Refinement** | |
| Resolution (Å) | 3.23 |
| $R_{work}$ | 0.265 |
| $R_{free}$ | 0.276 |
| RMS deviations | |
| Bond lengths (Å) | 0.003 |
| Bond angles (°) | 1.297 |
| **Validation** | |
| Molprobity score | 1.11 |
| Clashscore, all atoms | 1.07 |
| Favored rotamers (%) | 97.7 |
| Ramachandran plot (%) | |
| Favored | 95.7 |
| Allowed | 4.3 |
| Outliers | 0.0 |

functions (*Figure 5—figure supplement 1*), indicating a single slow step governing the OF lifetime. This is probably because ATP hydrolysis ($k_2$) is a fast step followed by a comparatively slow NBD dissociation step ($k_3$). The cryo-EM data further support that at physiological ATP concentrations (1–10 mM), the rate-limiting step is the dissociation of the NBD dimer subsequent to ATP hydrolysis (*Figure 7*). Recent studies of P-glycoprotein (*Bársony et al., 2016*), TmrAB (*Hofmann et al., 2019*), and TM287/288 (*Hutter et al., 2019*) also indicated that in their respective transport cycles, the OF-to-IF transition is rate limiting.

The cryo-EM structure of MRP1 shows that the ATP hydrolysis step per se does not induce any conformational change (*Figure 6*). One might then ask, what is the role of ATP hydrolysis in the transport cycle? Thermodynamically, the energy provided by ATP hydrolysis is necessary for uphill substrate translocation. Without energy dissipation, the system can only reach equilibrium, in which the substrate concentration is equal on both sides of the membrane. Kinetically, ATP hydrolysis provides directionality to the transport cycle: because $k_2$ is much greater than the reverse isomerization rate $k_{-1}$, estimated from the E1454Q cycling traces (*Figure 7—figure supplement 1*), the majority of molecules proceed to ATP hydrolysis upon NBD dimerization, followed by isomerization back to the IF conformation with a rate of $k_3$. Although the E1454Q mutant is catalytically deficient, it displayed transitions between IF and OF conformations with kinetics similar to the WT transporter (*Table 1*).

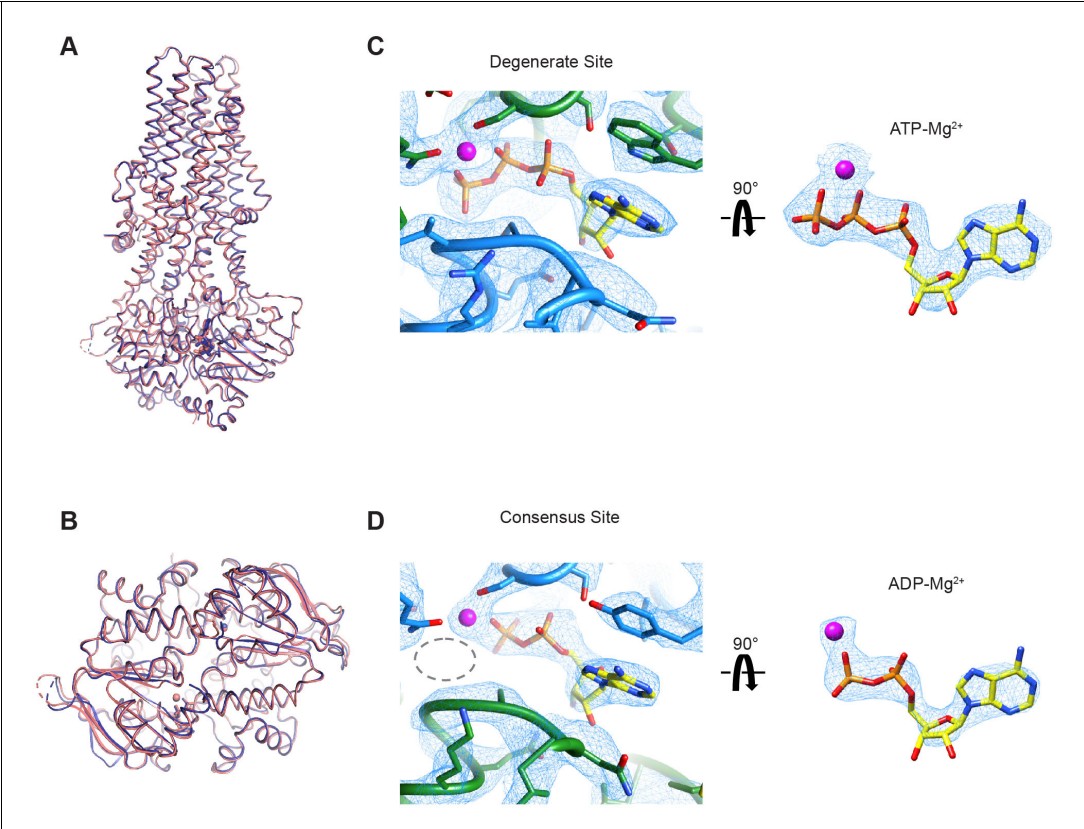

**Figure 6.** Cryo-EM structure of wild-type MRP1 in its post-hydrolytic state. (**A**) The structure of WT bMRP1 in the presence of saturating ATP and LTC$_4$ shown in blue, overlaid with the structure of the non-hydrolyzing bMRP1-E1454Q mutant in the presence of saturating ATP and LTC$_4$ (PDB 6BHU) shown in salmon. The structures are shown in cartoon representation viewed from within the plane of the membrane with ATP/ADP shown as sticks and Mg$^{2+}$ shown as spheres. (**B**) The same structural overlay as in (**A**), rotated 90° to view the NBD dimer from the cytoplasmic side. (**C**) Cryo-EM density for the degenerate ATPase site (left) and the ATP-Mg$^{2+}$ from the degenerate site alone (right, rotated 90°). NBD1 is shown in green, NBD2 in blue, ATP as yellow sticks colored by heteroatom, and Mg$^{2+}$ as a magenta sphere. (**D**) Cryo-EM density for the consensus ATPase site (left) and the ADP-Mg$^{2+}$ from the consensus site alone (right, rotated 90°). In the left panel, the position of the missing γ-phosphate is demarcated with a gray dotted oval. The color code is the same as in (**C**).

The online version of this article includes the following figure supplement(s) for figure 6:

**Figure supplement 1.** Cryo-EM data processing workflow.

**Figure supplement 2.** Quality of the cryo-EM maps.

This apparent paradox can be explained by our kinetic model (*Figure 7*): WT MRP1 predominantly takes the irreversible, hydrolysis-driven route ($k_1 \rightarrow k_2 \rightarrow k_3$), while the E1454Q mutant takes the reversible pathway ($k_1 \leftrightarrow k_{-1}$) that does not involve hydrolysis. The relative IF/OF occupancy is similar between WT and E1454Q MRP1 because $k_{-1}$ and $k_3$ share similar values, even though the E1454Q mutant cannot power uphill substrate transport.

When correlating smFRET and cryo-EM data, one must keep in mind that each technique has its own limitations and is carried out under very different experimental conditions. For example, multiple apo states were detected by smFRET but our previous cryo-EM study revealed only one ligand-free structure (*Johnson and Chen, 2017*). This discrepancy suggests that smFRET could be more sensitive in detecting sparsely populated species than cryo-EM. Furthermore, the FRET experiments were carried out at 23°C, whereas the cryo-EM structure was determined at cryogenic temperature. We do not fully understand how the conformational distribution of molecules on the EM grids might change upon rapid freezing. However, it seems possible that the occupancy of the lowest energy state may become more dominant as a result of a smaller $k_BT$ in the Boltzmann distribution.

The kinetics of another ABC protein, the cystic fibrosis transmembrane conductance regulator (CFTR), have been well characterized through functional studies (*Csanády et al., 2019*). CFTR is

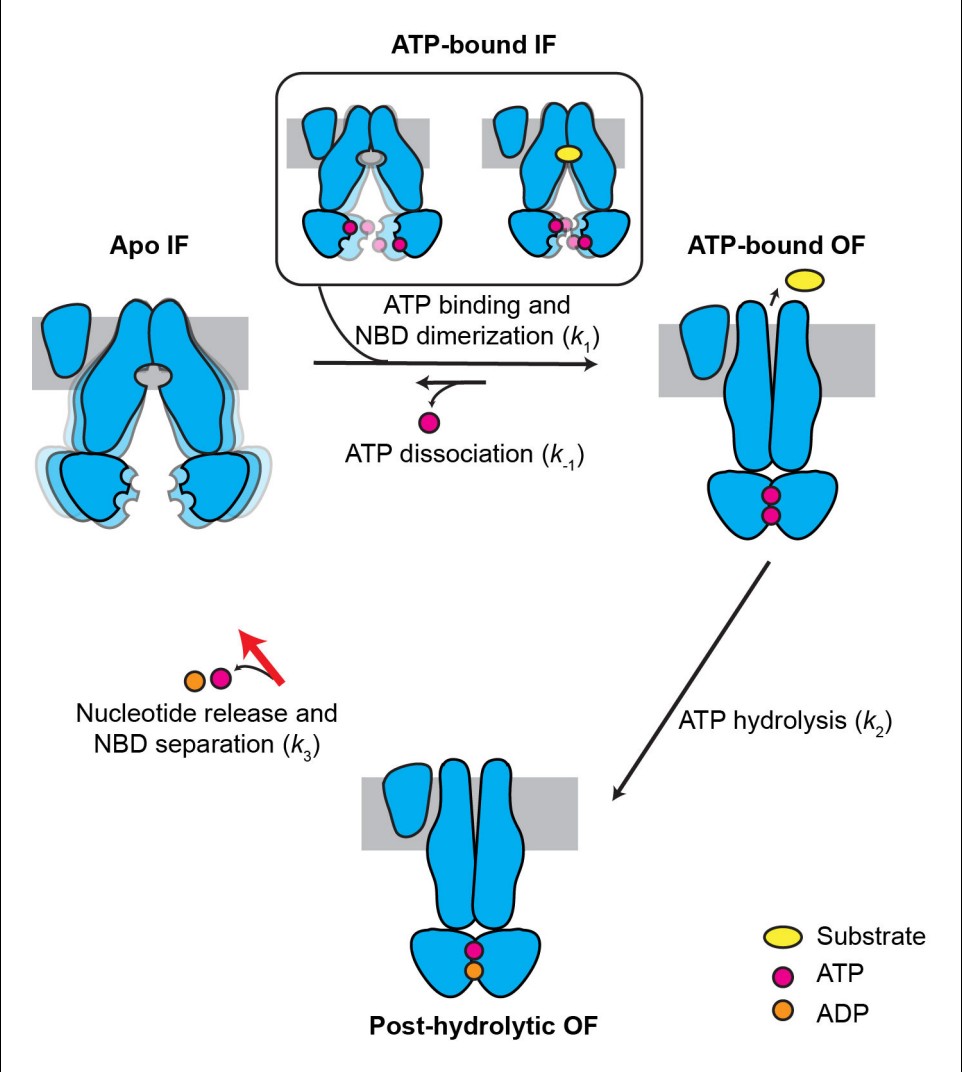

**Figure 7.** Kinetic model for the transport cycle of MRP1. MRP1 is intrinsically dynamic, transitioning between multiple IF conformations both in the absence and presence of ATP. Under physiological conditions, ATP rapidly binds to the IF state, promoting NBD dimerization and formation of the OF state. $LTC_4$ accelerates the IF-to-OF transition ($k_1$) but not the other transitions, yielding a faster overall ATPase turnover rate. The reverse isomerization ($k_{-1}$) resulting from ATP dissociation occurs at a much slower rate than the forward reaction. ATP hydrolysis in the consensus site ($k_2$) is fast and results in an asymmetric post-hydrolytic OF state with ATP in the degenerate site and ADP in the consensus site. This represents the predominant conformation determined by cryo-EM under active turnover conditions. The entire transport cycle is limited by the rate of dissociation of the NBD dimer ($k_3$) after ATP hydrolysis.

The online version of this article includes the following figure supplement(s) for figure 7:

**Figure supplement 1.** Perturbation experiments with E1454Q mutant MRP1.

closely related to MRP1 as both belong to the ABCC subfamily of ABC transporters. However, it functions as an ion channel instead of an active transporter and is thus considered to be a 'broken' ABC transporter. The gating cycle of CFTR is generally similar to the transport cycle of MRP1: ATP binding stabilizes an NBD-dimerized conformation in which the pore is open to allow ion conduction, and ATP hydrolysis is followed by NBD separation and channel closure. However, the kinetic properties of CFTR, characterized through single-channel recordings, are markedly different from those of MRP1. In the CFTR gating cycle, the rate-limiting step is the formation of the NBD-dimerized conformation (equivalent to $k_1$ in *Figure 7*), and ATP hydrolysis occurs much slower than NBD dissociation (*Csanády et al., 2010*; *Vergani et al., 2003*). Do these differences reflect the general distinction

between an active transporter and a passive channel? The answer awaits detailed kinetic characterization of other ABC transporters.

# Materials and methods

**Key resources table**

| Reagent type (species) or resource | Designation | Source or reference | Identifiers | Additional information |
|---|---|---|---|---|
| Antibody | Biotinylated 6x-His Tag monoclonal antibody | Invitrogen | Cat# MA1-21315-BTIN | Molar ratio of 2:1 (antibody:bMRP1) |
| Cell line | Sf9 | ATCC | CRL-1711 | |
| Cell line | HEK293S GnTI⁻ | ATCC | CRL-3022 | |
| Recombinant DNA reagent | bovine MRP1 in pUC57 vector | Bio Basic | | Codon-optimized |
| Recombinant DNA reagent | bovine MRP1 in a modified pEG BacMam vector | *Johnson and Chen, 2017* | | Suitable for expression in mammalian cells |
| Recombinant DNA reagent | bovine MRP1 with S6/A1 peptides for site-specific labeling | This paper | | |
| Recombinant DNA reagent | bovine MRP1 E1454Q with S6/A1 peptides for site-specific labeling | This paper | | |
| Recombinant DNA reagent | Sfp pet29b C-terminal His Tag | (*Worthington and Burkart, 2006*) | Addgene Plasmid# 75015 | |
| Recombinant DNA reagent | pET15b-ACPs (from *S. pneumoniae*) | Gift from Michael Johnson | Addgene Plasmid# 63687 | |
| Chemical compound, drug | Cy3 maleimide mono-reactive dye | GE Healthcare | Cat# PA23031 | |
| Chemical compound, drug | LD655 maleimide mono-reactive dye | Lumidyne Technologies | Cat# LD655-MAL | |
| Chemical compound, drug | Trolox | Sigma-Aldrich | Cat# 238813 | |
| Chemical compound, drug | 4-Nitrobenzyl alcohol (NBA) | Sigma-Aldrich | Cat# N12821 | |
| Chemical compound, drug | Cyclooctatetraene (COT) | Sigma-Aldrich | Cat# 138924 | |
| Chemical compound, drug | 3,4-Dihydroxy benzoic acid (PCA) | Sigma-Aldrich | Cat# 37580 | |
| Chemical compound, drug | Digitonin | Sigma-Aldrich | Cat# D141 | |
| Peptide, recombinant protein | Protocatechuate 3, 4-Dioxygenase (PCD) | Sigma-Aldrich | Cat# P8279 | |
| Peptide, recombinant protein | Leukotriene C4 | Cayman Chemical | Cat# 20210 | |
| Commercial assay, kit | NHS-activated Sepharose 4 Fast Flow resin | GE Healthcare | Cat# 17-0430-01 | |
| Commercial assay, kit | Superose 6, 10/300 GL | GE Healthcare | Cat# 17-5172-01 | |
| Software, algorithm | SPARTAN | (*Juette et al., 2016*) | https://www.scottcblanchardlab.com/software | |
| Software, algorithm | ebFRET | (*van de Meent et al., 2014*) | http://ebfret.github.io | |
| Software, algorithm | MATLAB | MathWorks | https://www.mathworks.com/products/matlab.html | |
| Software, algorithm | Origin | OriginLab | https://www.originlab.com | |

*Continued on next page*

*Continued*

| Reagent type (species) or resource | Designation | Source or reference | Identifiers | Additional information |
|---|---|---|---|---|
| Software, algorithm | GraphPad Prism | GraphPad | https://www.graphpad.com/scientific-software/prism/ | |
| Software, algorithm | RELION 1.4 | (*Scheres, 2012*) | https://www2.mrc-lmb.cam.ac.uk/relion | |
| Software, algorithm | Frealign | (*Grigorieff, 2016*) | https://grigoriefflab.janelia.org/frealign | |
| Software, algorithm | Coot | (*Emsley and Cowtan, 2004*) | https://www2.mrc-lmb.cam.ac.uk/personal/pemsley/coot | |
| Software, algorithm | PHENIX | (*Adams et al., 2010*) | https://www.phenix-online.org | |
| Software, algorithm | REFMAC | (*Brown et al., 2015*) | https://www.ccp4.ac.uk/html/refmac5.html | |
| Software, algorithm | MolProbity | (*Chen et al., 2010*) | https://molprobity.biochem.duke.edu | |
| Software, algorithm | Chimera | (*Pettersen et al., 2004*) | https://www.cgl.ucsf.edu/chimera | |
| Software, algorithm | PyMOL | PyMOL | https://www.pymol.org | |
| Software, algorithm | cryoSPARC | (*Punjani et al., 2017*) | https://cryosparc.com | |
| Other | R1.2/1.3 400 mesh Au holey carbon grids | Quantifoil | Cat# 1210627 | 1 µg/mL |

## Cell culture

Sf9 cells were cultured in sf-900 II SFM medium (GIBCO) supplemented with 5% FBS at 27°C. HEK293S GnTI⁻ suspension cells were cultured in Freestyle 293 medium (GIBCO) supplemented with 2% FBS at 37°C with 8% $CO_2$ and 80% humidity.

## Protein expression, purification, and site-specific labeling

All bMRP1 constructs were expressed and purified as described previously (*Johnson and Chen, 2017*). Briefly, baculovirus with each bMRP1 construct was generated and used to infect HEK293S GntI⁻ suspension cells. All constructs contained a C-terminal PreScission-Protease-cleavable GFP tag. Cell pellets were solubilized by adding 2% lauryl maltose neopentyl glycol (LMNG) supplemented with 0.2% cholesteryl hemisuccinate (CHS). After removal of the insoluble fraction by centrifugation, supernatants were batch bound to GFP-nanobody-conjugated Sepharose 4 Fast Flow resin (GE Healthcare). The resin was then packed into a column, washed with buffer containing 0.06% digitonin, and protein was eluted by digestion with PreScission Protease. After elution, protein was concentrated and applied to a Superose 6 10/300 GL column (GE Healthcare) equilibrated in 0.06% digitonin, 150 mM KCl, 50 mM Tris-HCl pH 8.0, 2 mM $MgCl_2$, and 2 mM DTT. Peak fractions were pooled and concentrated, and either used immediately to prepare cryo-EM samples or flash-frozen in liquid nitrogen and stored at −80°C for future FRET and ATPase assays.

For site-specific fluorescence labeling, the 12-residue S6 peptide (GDSLSWLLRLLN) was substituted at linker residues 868–879, and the 12-residue A1 peptide (GDSLDMLEWSLM) was added to the C-terminus immediately following V1530 (*Yin et al., 2006*; *Zhou et al., 2007*). A His$_{10}$-tag was also added at the N-terminus of bMRP1 for surface immobilization. Fluorescent labeling of the A1 site was carried out after elution off the GFP nanobody column by adding 5 µM AcpS, 25 µM LD655-CoA, and 10 mM $MgCl_2$ and incubating for 1 hr at room temperature. Excess dye was removed by size-exclusion chromatography as described above. The S6 site was then labeled by adding 5 µM Sfp, 25 µM Cy3-CoA, and 10 mM $MgCl_2$ and incubating for 1 hr at room temperature. The doubly labeled protein was cleaned up by size-exclusion chromatography. The labeling efficiencies were estimated to be ~30% for Cy3 and ~80% for LD655 from the respective extinction

coefficients of the protein and fluorophores. Sfp and AcpS synthases, as well as dye-CoA conjugates, were purified as described previously (*Yin et al., 2006*; *Zhou et al., 2007*).

## ATPase assay

ATP hydrolysis was monitored using an NADH-coupled ATPase assay (*Scharschmidt et al., 1979*). The reaction contained 800 nM bMRP1, 60 µg/mL pyruvate kinase, 32 µg/mL lactate dehydrogenase, 9 mM phosphoenolpyruvate, and 150 µM NADH in a buffer containing 50 mM Tris-HCl pH 8.0, 150 mM KCl, 2 mM $MgCl_2$, 2 mM DTT, and 0.06% digitonin. ATP-$Mg^{2+}$ was added to initiate the reaction and the consumption of NADH was measured at 30°C by monitoring the fluorescence at $\lambda_{ex}$ = 340 nm and $\lambda_{em}$ = 445 nm using an Infinite M1000 microplate reader (Tecan). $LTC_4$ stimulation experiments were performed by pre-incubating 10 µM $LTC_4$ with the reaction mix for 15 min before initiating the reaction.

## Single-molecule fluorescence imaging

Single-molecule experiments were performed at room temperature (23 ± 1°C) on an objective-type total-internal-reflection fluorescence microscope (Olympus IX83 cellTIRF). Microfluidic imaging chambers were passivated with a mixture of PEG and biotin-PEG (Laysan Bio), and incubated with 0.8 µM streptavidin (Invitrogen) followed by 2 nM fluorescently labeled, His-tagged bMRP1 that had been preincubated with biotinylated anti-$His_6$ antibodies (Invitrogen) for 1 hr on ice in a buffer containing 50 mM Tris-HCl pH 8.0, 150 mM KCl, 2 mM $MgCl_2$, 0.06% digitonin, 0.5 mg/mL BSA, 10 mM phosphocreatine (Sigma), and 0.1 mg/mL creatine kinase (Sigma). A triplet-state quenching cocktail (*Dave et al., 2009*) of 1 mM cyclooctatetraene (Sigma), 1 mM 4-nitrobenzyl alcohol (Sigma), and 1 mM Trolox (Sigma), as well as an oxygen scavenging system (*Aitken et al., 2008*) containing 10 nM protocatechuate-3,4-dioxygenase (Sigma) and 2.5 mM protocatechuic acid (Sigma) were supplemented to the imaging buffer. ATP and/or $LTC_4$ were included in the imaging buffer at concentrations specified in the text. Fluorescence signals were split with a W-View Gemini-2C (Hamamatsu), directed to two CMOS cameras (Flash 4.0 v3, Hamamatsu), and acquired by MetaMorph software (Molecular Devices) at a frame rate specified in the text.

## Analysis of smFRET data

Single-molecule fluorescence-time trajectories were extracted and subsequently analyzed using the SPARTAN software (*Juette et al., 2016*) in MATLAB (MathWorks). The FRET efficiency (*E*) was calculated as $I_A/(I_D+I_A)$, where $I_D$ and $I_A$ represent the donor and acceptor fluorescence intensities, respectively. Aberrant and/or noisy traces were filtered by applying the following criteria: single-step donor photobleaching, $SNR_{Background} \geq 8$, <4 donor blinking events, and FRET efficiency above 0.1 for at least 15 frames.

The remaining traces were idealized using hidden Markov modeling (HMM) (*Qin, 2004*) implemented in SPARTAN. Initial model parameters were obtained by running ebFRET (*van de Meent et al., 2014*) with single-molecule trajectories for WT MRP1 with 5 mM ATP and 10 µM $LTC_4$ collected at 25 ms resolution to ensure that all functionally relevant states were readily sampled. Potential models containing two to six non-zero FRET states with initial model parameters generated from ebFRET were applied to raw FRET trajectories from each condition using the segmental *k*-means algorithm (*Qin, 2004*) in SPARTAN, yielding optimized model parameters and idealized trajectories. A five-state model was chosen based on the following: (1) the lower bound evidence from ebFRET; (2) FRET state assignment histograms showing populated, symmetric peaks centered at each model-assigned FRET value; and (3) visual inspection of idealized trajectories from models with 2–6 non-zero FRET states. A set of five FRET states with mean *E* values of 0.23, 0.42, 0.63, 0.80, and 0.92 was finally chosen and fixed to describe data across all conditions.

HMM analysis of the smFRET trajectories identified transitions between each idealized state and was subsequently used to construct the corresponding transition density plots (*McKinney et al., 2006*). Well-separated peaks in the transition density plots lent further support for a five-state model (*Figure 3—figure supplement 2*). FRET contour plots and histograms were built from the first 50 or 200 frames of each trajectory with a bin size of 0.03 and plotted in Origin (OriginLab). Occupancies of each state were plotted with GraphPad Prism 7.

## Kinetic analysis of smFRET data

HMM analysis yielded the dwell times in each IF state from the idealized traces collected at a 25 ms frame rate. With this imaging condition, the effective FRET observation window is 4.1 ± 0.5 s limited by dye photobleaching. At a 300 ms frame rate, the observation window increased to 101 ± 13 s due to lower laser intensities required to achieve the same signal-to-noise ratio. However, the lower time resolution obscured fast transitions among IF states. Therefore, we grouped the four IF states ($IF_1$, $IF_2$, $IF_3$, and $IF_4$) as a composite IF state and extracted the time that molecules spent in the IF ($t_{IF}$) and OF ($t_{OF}$) state from active molecules taken with a 300 ms frame rate. Active molecules were defined as those that successfully transitioned to the OF state after ATP injection and represent ~60% of the whole population. The remaining molecules (~40%) never visited the OF state before photobleaching, thus were assigned as the inactive group. The $t_{OF}$ histograms were fit by single-exponential functions, yielding characteristic decay constants $\tau_{OF}$. Mean values of $t_{IF}$ were used to describe the lifetimes of the composite IF state. These values are reported after correction for dye photobleaching ($k_{photobleaching}$ = 0.010 ± 0.001 s$^{-1}$). The reported errors represent the propagated SEM from the observed IF/OF lifetimes and fluorophore lifetimes.

## Cryo-EM sample preparation and data collection

Purified WT bMRP1 was mixed with 10 mM ATP and 80 µM $LTC_4$ in DMSO (2.5% final concentration) and incubated on ice for 10 min. Immediately before freezing grids, 3 mM fluorinated Fos-choline-8 was added, yielding a final protein concentration of 5.3 mg/mL. Sample was applied to freshly glow-discharged Quantifoil R1.2/1.3 400-mesh Au Holey Carbon Grids and frozen in liquid ethane using a Vitrobot Mark IV (FEI).

Cryo-EM data were collected using a Titan Krios system (FEI) with a K2 Summit camera (Gatan) in super resolution mode at a pixel size of 0.515 Å/pixel. The electron dose rate was eight electrons/pixel/sec for an exposure time of 10 s divided into 50 frames. A total of 3993 movies were collected (*Table 2*).

## Cryo-EM image processing, model building, and refinement

Movie frames were corrected for gain reference and binned by two to yield a pixel size of 1.03 Å/pixel. Sub-frame alignment was carried out using MotionCor2, and the contrast transfer function (CTF) was estimated using Gctf (*Zhang, 2016*; *Zheng et al., 2017*). From 3604 micrographs, 1,143,729 particles were selected using Gautomatch (http://www.mrc-lmb.cam.ac.uk/kzhang/). 2D classification was carried out in RELION (*Zivanov et al., 2018*), and the best class averages contained 644,840 particles. Nearly all 2D class averages appeared to be in the outward-facing, NBD-dimerized conformation (*Figure 6—figure supplement 1*). 3D classification with four classes was performed in RELION using the OF bMRP1-E1454Q map low-pass-filtered to 60 Å as a reference model. All four classes appeared to contain a closed NBD dimer. The best class of 257,107 particles was then subjected to a two-stage masked 3D refinement (initiated with a mask that included the micelle and entire protein and continued with a mask that excluded the micelle and TMD0), yielding a map at 3.8 Å. Iterative cycles of CTF refinement, Bayesian polishing, and masked 3D refinement in RELION were then performed. After several cycles of this process, masked 3D classification without alignment (using the angles from the most recent 3D refinement) was performed, yielding a best class of 81,078 particles. This subset was further subjected to iterative cycles of CTF refinement, Bayesian polishing, and masked 3D refinement in RELION, yielding a final map at 3.2 Å (*Figure 6— figure supplement 1*).

To generate a reconstruction using all 1,143,729 particles, refinement was performed in Frealign (*Grigorieff, 2016*) using the map generated in the first stage of RELION refinement as the reference model. A global search was first performed using information to 8 Å, followed by several rounds of local search.

Model building and refinement were carried out as previously described (*Johnson and Chen, 2018*). The OF E1454Q model (PDB 6BHU) was rigid-body fit into the map using UCSF Chimera (*Pettersen et al., 2004*) and real-space refined in PHENIX (*Adams et al., 2010*). The model was then subjected to iterative cycles of refinement in Refmac (*Brown et al., 2015*) and manual rebuilding in COOT (*Emsley and Cowtan, 2004*). MolProbity (*Chen et al., 2010*) was used to assess the quality of the final model (*Figure 6—figure supplement 2* and *Table 2*). The final model contains

residues 203–268, 311–635, 641–870, and 942–1530, as well as one molecule of ATP, one molecule of ADP, two $Mg^{2+}$ ions, and three partial cholesteryl hemisuccinate molecules. $R_{work}$ and $R_{free}$ values were calculated by generating a mask from the model with 2 Å padding, applying it to each of the half maps, and running zero cycles of refinement in Refmac against the working map (half-map 1, $R_{work}$) and the free map (half-map 2, $R_{free}$). Figures were prepared using UCSF Chimera and PyMOL (Schrödinger, LLC).

## Statistical analysis

Comparisons between conditions were made in GraphPad Prism seven using unpaired two-tailed Student's *t*-tests unless specified otherwise. A threshold of $p<0.05$ was chosen to determine statistical significance (*$p<0.05$; **$p<0.01$; ***$p<0.001$; ****$p<0.0001$; ns, not significant). The number of single molecules analyzed is indicated in the figure panels. Data for each condition were collected in multiple batches across different days. Unless otherwise noted, errors reported in this study represent the approximated standard error of the mean determined from 10,000 bootstrapped samples.

## Acknowledgements

We thank Michael L Oldham for help with cryo-EM data processing, Roderick MacKinnon for comments, and staff at Rockefeller University's Evelyn Gruss Lipper Cryo-Electron Microscopy Resource Center for assistance with data collection. LW is a Rockefeller University Women and Science postdoctoral fellow. MRW is a Rockefeller University Anderson Cancer Center postdoctoral fellow. ZLJ is a fellow of the Jane Coffin Childs Memorial Fund for Medical Research. SL is supported by the Robertson Foundation, a Monique Weill-Caulier Career Award, a Kimmel Scholar Award, a Sinsheimer Scholar Award, and an NIH Director's New Innovator Award (DP2HG010510). JC is an investigator of the Howard Hughes Medical Institute.

## Additional information

### Funding

| Funder | Grant reference number | Author |
| --- | --- | --- |
| NIH Office of the Director | DP2HG010510 | Shixin Liu |
| Howard Hughes Medical Institute | Investigatorship | Jue Chen |
| Rockefeller University | Women and Science postdoctoral fellow | Ling Wang |
| Rockefeller University | Anderson Cancer Center postdoctoral fellow | Michael R Wasserman |
| Jane Coffin Childs Memorial Fund for Medical Research | | Zachary Lee Johnson |
| Robertson Foundation | | Shixin Liu |
| Monique Weill-Caulier Career Award | | Shixin Liu |
| Sidney Kimmel Foundation | Kimmel Scholar Award | Shixin Liu |
| SinsheimerScholar Award | | Shixin Liu |

The funders had no role in study design, data collection and interpretation, or the decision to submit the work for publication.

### Author contributions

Ling Wang, Zachary Lee Johnson, Michael R Wasserman, Data curation, Formal analysis, Investigation, Writing - original draft, Writing - review and editing; Jesper Levring, Data curation, Formal analysis, Investigation, Writing - review and editing; Jue Chen, Shixin Liu, Conceptualization, Formal analysis, Supervision, Funding acquisition, Writing - original draft, Writing - review and editing

Author ORCIDs
Jue Chen ![ORCID] https://orcid.org/0000-0003-2075-4283
Shixin Liu ![ORCID] https://orcid.org/0000-0003-4238-7066

Decision letter and Author response
Decision letter https://doi.org/10.7554/eLife.56451.sa1
Author response https://doi.org/10.7554/eLife.56451.sa2

## Additional files

### Supplementary files
- Transparent reporting form

### Data availability

The cryo-EM density map for bMRP1 in its post-hydrolytic state is deposited in the Electron Microscopy Data Bank (EMDB) under accession code EMD-20945. The coordinates for the model are deposited in the Protein Data Bank (PDB) under accession number 6UY0.

The following datasets were generated:

| Author(s) | Year | Dataset title | Dataset URL | Database and Identifier |
|---|---|---|---|---|
| Johnson ZL, Chen J | 2020 | Cryo-EM structure of wild-type bovine multidrug resistance protein 1 (MRP1) under active turnover conditions | http://www.rcsb.org/structure/6UY0 | RCSB Protein Data Bank, 6UY0 |
| Johnson ZL, Chen J | 2020 | Cryo-EM structure of wild-type bovine multidrug resistance protein 1 (MRP1) under active turnover conditions | https://www.ebi.ac.uk/pdbe/entry/emdb/EMD-20945 | Electron Microscopy Data Bank, EMD-20945 |

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
