## [Decision Letter]

**Acceptance summary:**

In this pioneering study the authors use single-molecule fluorescence spectroscopy to track the conformational changes of the asymmetric ABC transporter bovine Multidrug resistance protein 1 (MRP1) in real time, and obtain a cryo-EM structure of MRP1 under active turnover conditions. The experiments show that MRP1 spends most of its time in the outward-facing (OF) conformation during its catalytic cycle. Correspondingly, the majority of the particles analyzed for cryo-EM are in an outward-facing (OF) conformation, although the ATP molecule bound at the active catalytic site has been hydrolyzed. The study provides two major functional conclusions: (i) substrate stimulates ATP hydrolysis by accelerating the IF-to-OF transition, and (ii) the rate-limiting step of the transport cycle is the dissociation of the nucleotide-binding-domain dimer, not ATP hydrolysis per se, as has been generally assumed. The paper is of broad general interest to the entire ABC protein field, and a pioneering example of exploiting optical methods for real-time single-molecule biophysics.

**Decision letter after peer review:**

Thank you for submitting your article "Characterization of the kinetic cycle of an ABC transporter by single-molecule and cryo-EM analyses" for consideration by *eLife*. Your article has been reviewed by three peer reviewers, including László Csanády as the Reviewing Editor and Reviewer #1, and the evaluation has been overseen by Olga Boudker as the Senior Editor. The following individuals involved in review of your submission have agreed to reveal their identity: Markus A Seeger (Reviewer #2); Gergely Szakacs (Reviewer #3).

The reviewers have discussed the reviews with one another and the Reviewing Editor has drafted this decision to help you prepare a revised submission. In recognition of the fact that revisions may take longer than we typically allow, until the research enterprise restarts in full, we will give authors as much time as they need to submit revised manuscripts.

Summary:

This is a highly interesting study in which the authors use single-molecule fluorescence spectroscopy to track the conformational changes of the asymmetric ABC transporter bovine Multidrug resistance protein 1 (MRP1) in real time. Using Bayesian methods they identify five distinct molecular conformations of the transporter, and quantitate the life times spent in each of these states, as well as their dependences on ATP and substrate concentrations. In addition, the authors obtain a cryo-EM structure of MRP1 under active turnover conditions, and find that the majority of the particles is in an outward-facing (OF) conformation, although the ATP molecule bound at the active catalytic site has been hydrolyzed. The study provides two major functional conclusions: (i) substrate stimulates ATP hydrolysis by accelerating the IF-to-OF transition, and (ii) the rate-limiting step of the transport cycle is the dissociation of the nucleotide-binding-domain dimer, not ATP hydrolysis per se, as has been generally assumed. The paper is of broad general interest to the entire ABC protein field, and a pioneering example of exploiting optical methods for real-time single-molecule biophysics.

The reviewers were all enthusiastic about the unique combination of approaches used, and the potential that these techniques offer for deciphering mechanistic details of a transporter cycle at a molecular level. On the other hand, several important concerns were raised regarding the choice of experimental conditions, and the interpretation of the data in terms of a kinetic model. These should be addressed by the authors.

Essential revisions:

1) The authors claim that the cryo-EM structure was obtained under active turnover conditions. On the other hand, the Materials and methods section (subsection “Cryo-EM sample preparation and data collection”) states that the protein was incubated with ATP and LTC_4_ for 10 minutes on ice, before flash-freezing. Considering the turnover rate at room temperature (Figure 1C, temperature not specified, but we assume it was RT), 10 minutes on ice may not allow the completion of the cycle, hence, it is questionable whether "active turnover conditions" were met. Furthermore, since the functional (FRET and ATPase) measurements were done at RT, a direct comparison of the structural and functional data is problematic (the various kinetic steps in the cycle are likely to have different temperature dependences, thus, the fractional times spent in IF and OF states are also most likely temperature dependent). The authors should clarify this important technical caveat.

2) There are several major inconsistencies among the kinetic parameters extracted from the functional data. These require a major re-thinking of the quantitative analysis. We can envisage two possible acceptable outcomes: (i) construction of a kinetic scheme that at least semi-quantitatively accounts for both the presented structural (cryo-EM) and functional (FRET and ATPase) data, or (ii) toning down the mechanistic conclusions and clearly spelling out apparent inconsistencies between the various data sets, pointing out that clarification of these will require further work. Below are some of the inconsistencies noted:

2.1) Figure 2A: For E1454Q in the presence of ATP+LTC_4_ the relative occupancy of the OF state (fractional area under the magenta peak) is only ~40%. This seems inconsistent with the fact that under the same conditions a cryo-EM study on this mutant (Johnson and Chen, 2018) revealed a single dominant population of molecules in the OF conformation. The ~40% fractional occupancy of the OF state for E1454Q (Figure 2A) is also not consistent with the life times of the IF and OF states for this mutant, listed in Table 1: with t(IF)=7.7 s and t(OF)=31.7 s the mutant should spend 80% of its time in OF (i.e., at any point in time 80% of the population should be in OF).

Along the same lines, under conditions that allow catalytic turnover, all MRP1 molecules should be trapped with vanadate, whereas the FRET histograms for Vi/ATP/LTC_4_ in Figure 2A also indicate the presence of several IF states.

How do the authors explain the prevalence of IF conformations in these conditions (E1454Q and WT+ATP+Vi)? Can the authors rule out that (at least some of) the IF1-4 conformations shown in Figure 2A represent either artifacts (wobbling of attached fluorophores?), or protein conformations that are not part of the physiological catalytic cycle? E.g., several studies have shown that a nucleotide is stably bound to NBD1. Do the authors assume that MRP1 adopts the IF conformations devoid of ATP in a physiologically relevant setting?

2.2) Based on Table 1, for FRET-labeled WT MRP1 the cycle time in the presence of saturating ATP and LTC_4_ is T(cycle)=t(IF)+t(OF)=4.7s + 30.8 s = 35.5 s, which translates into a kcat of ~1.7/min. How do the authors reconcile these data with the kcat of ~11/min reported under identical conditions for FRET-labeled WT in Figure 1C?

2.3) There is a discrepancy between the effects of LTC_4_ on kcat (Figure 1C) and IF/OF dwell times (Table 1). Based on Table 1, for WT MRP1 the total cycle time (T(cycle)=t(IF)+t(OF)) is 44.9 s in saturating ATP, but 35.5 s in saturating ATP+LTC_4_. This predicts only ~1.26x stimulation of kcat by LTC_4_, whereas a >2x increase is reported in Figure 1C. (And the absolute kcat values calculated from the cycle times are much smaller than those in Figure 1C, see comment 2.2., above.)

3) Throughout the manuscript, the E1454Q mutant is described as hydrolysis-deficient. However, in Figure 1C the activity of the mutant is only ~3x slower than WT in the absence, and ~10x slower than WT in the presence, of LTC_4_. How do the authors interpret this background ATPase activity? Does it reflect residual activity of the mutant or a minute contamination by a highly active ATPase? Please discuss.

4) The transition patterns shown in all four panels of Figure 3B appear highly symmetrical with respect to the y=x diagonal, even for the panels with ATP and ATP+LTC_4_. This suggests that the conformational changes are at equilibrium: there is no observable time-asymmetry. On the other hand, one would expect pronounced time asymmetry for the conformational transitions of an active transporter undergoing a unidirectional, non-equilibrium cycle associated with ATP hydrolysis. One possibility is that – with the fluorophores inserted into their current positions – FRET efficiency does not differentiate between functionally distinct states (e.g., between ATP/ATP-bound IF (pre-OF) and ATP/ADP-bound IF (post-OF), or between ATP/ATP-bound OF (pre-hydrolysis) and ATP/ADP-bound OF (post-hydrolysis)). Would it be feasible to introduce the FRET labels into a pair of positions that track motions around the active ATPase site more sensitively, so that, e.g., pre- and posthydrolytic OF states could be resolved? Alternatively, as the fractional occupancy of an intermediate is inversely correlated with its free energy, some functionally relevant states might have such short life times that they are not resolved by the assay. Pointing out these issues would provide further depth to the Discussion.

5) Please cite some earlier studies on other ABC proteins, that have arrived at similar conclusions. The conclusion that in MRP1 ATP hydrolysis occurs much faster than NBD separation is reminiscent to earlier observations on the bacterial ABC exporter TM287/288 (Hutter et al., 2019, PMID: 31113958). The conclusion that under active turnover conditions MRP1 spends the majority of its time in OF (i.e., the IF->OF transition is not rate limiting) is reminiscent to earlier observations on PgP (Barsony et al., 2016, PMID: 27117502).

---

## [Author Response]

Essential revisions:1) The authors claim that the cryo-EM structure was obtained under active turnover conditions. On the other hand, the Materials and methods section (subsection “Cryo-EM sample preparation and data collection”) states that the protein was incubated with ATP and LTC_4_ for 10 minutes on ice, before flash-freezing. Considering the turnover rate at room temperature (Figure 1C, temperature not specified, but we assume it was RT), 10 minutes on ice may not allow the completion of the cycle, hence, it is questionable whether "active turnover conditions" were met.

We have now clearly defined the “active turnover” conditions in the text: “To preparethesample for cryo-EM, WT bMRP1 (30 µM) was incubated with80 µM LTC_4_and10 mM ATP-Mg^2+^on ice for 10 minutes before plunge freezingin liquid ethane.[…]This approximates an‘active turnover’ conditionwhere ATP hydrolysishasreached steady state”. Our rationale for incubating at the lower temperature was to allow a sufficient amount of time for nucleotide binding and at least one round of ATP hydrolysis for every bMRP1 molecule while maintaining a relatively steady concentration of ATP/ADP. The fact that the majority of particles in the cryo-EM sample displayed an ADP molecule in the consensus ATPase site indicates that most molecules have completed at least one cycle of ATP hydrolysis.

In addition, the temperature of the ATPase measurement is now specified in the Materials and methods subsection “ATPase assay”.

Furthermore, since the functional (FRET and ATPase) measurements were done at RT, a direct comparison of the structural and functional data is problematic (the various kinetic steps in the cycle are likely to have different temperature dependences, thus, the fractional times spent in IF and OF states are also most likely temperature dependent). The authors should clarify this important technical caveat.

We agree. We have now discussed this extensively in the revised manuscript (Discussion, fifth paragraph).

2) There are several major inconsistencies among the kinetic parameters extracted from the functional data. These require a major re-thinking of the quantitative analysis. We can envisage two possible acceptable outcomes: (i) construction of a kinetic scheme that at least semi-quantitatively accounts for both the presented structural (cryo-EM) and functional (FRET and ATPase) data, or (ii) toning down the mechanistic conclusions and clearly spelling out apparent inconsistencies between the various data sets, pointing out that clarification of these will require further work. Below are some of the inconsistencies noted:

Thank you for pointing this out. Because of the inherent difference between the cryo-EM and smFRET experimental conditions, we refrained from making a quantitative assertion of the kinetic scheme. However, our major conclusion, the transport cycle is rate-limited by the OF-to-IF transition, is well supported by both smFRET and cryoEM data presented in this manuscript.

2.1) Figure 2A: For E1454Q in the presence of ATP+LTC_4_ the relative occupancy of the OF state (fractional area under the magenta peak) is only ~40%. This seems inconsistent with the fact that under the same conditions a cryo-EM study on this mutant (Johnson and Chen, 2018) revealed a single dominant population of molecules in the OF conformation. The ~40% fractional occupancy of the OF state for E1454Q (Figure 2A) is also not consistent with the life times of the IF and OF states for this mutant, listed in Table 1: with t(IF)=7.7 s and t(OF)=31.7 s the mutant should spend 80% of its time in OF (i.e., at any point in time 80% of the population should be in OF).Along the same lines, under conditions that allow catalytic turnover, all MRP1 molecules should be trapped with vanadate, whereas the FRET histograms for Vi/ATP/LTC_4_ in Figure 2A also indicate the presence of several IF states.How do the authors explain the prevalence of IF conformations in these conditions (E1454Q and WT+ATP+Vi)? Can the authors rule out that (at least some of) the IF1-4 conformations shown in Figure 2A represent either artifacts (wobbling of attached fluorophores?), or protein conformations that are not part of the physiological catalytic cycle? E.g., several studies have shown that a nucleotide is stably bound to NBD1. Do the authors assume that MRP1 adopts the IF conformations devoid of ATP in a physiologically relevant setting?

The reviewers are correct that some IF conformations come from molecules that are inactive, possibly caused by surface immobilization. The histograms presented in Figure 2A are from a mixture of active and inactive molecules. The lifetimes listed in Table 1 are calculated from active molecules only. In our initial submission, the existence of this inactive population was briefly described in the Materials and methods. In the revision, we now include detailed analysis of the active versus inactive molecules in the main text. Please see the second paragraph of the subsection “Determinants of IF-to-OF transitions” and the accompanying Figure 4—figure supplement 1.

Regarding the functional relevance of the four IF states identified from the smFRET data, we show that IF_1_ and IF_2_ can directly transition to the OF state (Figure 4—figure supplement 2A) and that IF_3_ and IF_4_ are also visited by active molecules under ATP turnover conditions (Figure 4—figure supplement 1). These multiple FRET states could reflect either global TMD movements or local NBD mobility. It is less likely that the FRET changes are due to wobbling of the attached fluorophores, because the rate of solvent-accessible fluorophore tumbling is orders of magnitude faster than our imaging time resolution and would thus be averaged out during each frame acquisition (PMID: 25368432). Therefore, we believe that these IF states correspond to authentic protein conformations sampled during active transport rather than artifacts. Whether each of these states – especially IF_3_ and IF_4_ – are functionally important requires further investigation. We now discuss this point in the second paragraph of the Discussion.

2.2) Based on Table 1, for FRET-labeled WT MRP1 the cycle time in the presence of saturating ATP and LTC_4_ is T(cycle)=t(IF)+t(OF)=4.7s + 30.8 s = 35.5 s, which translates into a kcat of ~1.7/min. How do the authors reconcile these data with the kcat of ~11/min reported under identical conditions for FRET-labeled WT in Figure 1C?

There are several factors that could contribute to the discrepancy between the ATP turnover rate determined from the single-molecule vs. bulk assay, now discussed in the subsection “Determinants of OF-to-IF transitions”.

1) The bulk ATPase assay was performed at 30 °C, while the smFRET data were collected at 23 °C.

2) The tethering of the bMRP1 protein to the slide surface in the smFRET assay may affect its activity to some extent. It is not uncommon that surface immobilization has an effect on protein dynamics in single-molecule studies. Nevertheless, the sensitivity displayed by surface-tethered MRP1 to ATP and substrate titrations (Figure 2—figure supplement 2) argues that the protein is still in its functional form.

3) The finite time resolution of the smFRET assay entails that the fastest fraction of IF-OF transitions may escape detection, making the apparent total cycle time longer than the true value. However, given the 300-ms frame rate and the average IF/OF lifetime (4.7 s for IF and 30.8 s for OF with saturating ATP and LTC_4_), we expect this effect to be minor.

2.3) There is a discrepancy between the effects of LTC_4_ on kcat (Figure 1C) and IF/OF dwell times (Table 1). Based on Table 1, for WT MRP1 the total cycle time (T(cycle)=t(IF)+t(OF)) is 44.9 s in saturating ATP, but 35.5 s in saturating ATP+LTC_4_. This predicts only ~1.26x stimulation of kcat by LTC_4_, whereas a >2x increase is reported in Figure 1C. (And the absolute kcat values calculated from the cycle times are much smaller than those in Figure 1C, see comment 2.2., above.)

This discrepancy may be explained by the same factors listed above in response to point #2.2. We now discuss the discrepancy between the smFRET and bulk ATPase assays in the revised manuscript (in the subsection “Determinants of OF-to-IF transitions”). We also refrained from making quantitative conclusions on the kinetic scheme but focused on general characteristics of the transport cycle.

3) Throughout the manuscript, the E1454Q mutant is described as hydrolysis-deficient. However, in Figure 1C the activity of the mutant is only ~3x slower than WT in the absence, and ~10x slower than WT in the presence, of LTC_4_. How do the authors interpret this background ATPase activity? Does it reflect residual activity of the mutant or a minute contamination by a highly active ATPase? Please discuss.

Because the background ATPase activity seen for the E1454Q mutant did not respond to LTC_4_ stimulation, we attribute it to spontaneous ATP hydrolysis – independent of bMRP1 activity – that is always observed in this type of assay from our experience. We have now added this clarification to the legend of Figure 1.

4) The transition patterns shown in all four panels of Figure 3B appear highly symmetrical with respect to the y=x diagonal, even for the panels with ATP and ATP+LTC_4_. This suggests that the conformational changes are at equilibrium: there is no observable time-asymmetry. On the other hand, one would expect pronounced time asymmetry for the conformational transitions of an active transporter undergoing a unidirectional, non-equilibrium cycle associated with ATP hydrolysis. One possibility is that – with the fluorophores inserted into their current positions – FRET efficiency does not differentiate between functionally distinct states (e.g., between ATP/ATP-bound IF (pre-OF) and ATP/ADP-bound IF (post-OF), or between ATP/ATP-bound OF (pre-hydrolysis) and ATP/ADP-bound OF (post-hydrolysis)). Would it be feasible to introduce the FRET labels into a pair of positions that track motions around the active ATPase site more sensitively, so that, e.g., pre- and posthydrolytic OF states could be resolved? Alternatively, as the fractional occupancy of an intermediate is inversely correlated with its free energy, some functionally relevant states might have such short life times that they are not resolved by the assay. Pointing out these issues would provide further depth to the Discussion.

We thank the reviewers for bringing up this point. The main purpose of the transition density plot (TDP) analysis was to validate the assignment of the four distinct IF states from the noisy single-molecule FRET traces: if the states are properly identified, they should appear as well-resolved peaks in the TDP and the transition pattern should be symmetrical with respect to the diagonal under equilibrium conditions (apo and +LTC_4_). This is the case as shown in the original Figure 3B, suggesting that the idealization of FRET traces appropriately resolved the discrete states. An additional benefit of the TDP analysis is that it yields the characteristic lifetimes of IF states.

As the reviewers pointed out, ATP drives the system out of equilibrium and should produce time asymmetry if the state path of the IF-to-OF transition differs from that of the OF-to-IF transition. However, because of the short lifetime of each IF state, the vast majority of transitions in the presence of ATP were still those among IF states. These fast transitions may have well approached equilibrium before the IF-to-OF isomerization. Because of the limited observation window at 25-ms time resolution and the long-lived OF states, the potentially asymmetrical, nonequilibrium transitions between IF and OF conformations make up only a small fraction of total transitions. Therefore, the TDPs, dominated by IF-IF transitions, still appear largely symmetrical even in the presence of ATP. This is further confounded by the aforementioned inactive population, which did not respond to ATP and likely remained at equilibrium. Unfortunately, it was not possible to distinguish between active and inactive molecules without the perturbation experiments conducted at 300-ms time resolution, which on the other hand obscured the fast transitions among IF states, and thus were unsuitable for TDP analysis.

We also agree with the other plausible explanation proposed by the reviewers for the apparent symmetrical TDP under nonequilibrium conditions, i.e. the smFRET assay did not resolve certain functionally distinct states that would break the symmetry. It is a common challenge in the single-molecule field to directly visualize transitions between pre- and post-hydrolytic states, especially when the catalytic step does not induce large-scale conformational rearrangements in the enzyme, as is the case for MRP1 (Figure 6). It may be possible to develop fluorophores that are sensitive to the chemical environment of the ATPase site. But such a resource is not yet widely available to our knowledge and would be outside the scope of this study. Moreover, given the finite time resolution of the assay, certain short-lived intermediates may have been missed. The fastest frame rate used in this work is 25 milliseconds, which was chosen to balance time-resolving power, signal-to-noise, and throughput.

Because of the uncertainties associated with the interpretation of TDPs for nonequilibrium conditions, we decide to only show TDPs for the equilibrium conditions (apo and +LTC_4_) as Figure 3—figure supplement 2 in the revised manuscript.

5) Please cite some earlier studies on other ABC proteins, that have arrived at similar conclusions. The conclusion that in MRP1 ATP hydrolysis occurs much faster than NBD separation is reminiscent to earlier observations on the bacterial ABC exporter TM287/288 (Hutter et al., 2019, PMID: 31113958). The conclusion that under active turnover conditions MRP1 spends the majority of its time in OF (i.e., the IF->OF transition is not rate limiting) is reminiscent to earlier observations on PgP (Barsony et al., 2016, PMID: 27117502).

We revised the manuscript as suggested (Discussion, third paragraph).